# Effect of Reactive Oxygen Scavenger N,N′-Dimethylthiourea (DMTU) on Seed Germination and Radicle Elongation of Maize

**DOI:** 10.3390/ijms242115557

**Published:** 2023-10-25

**Authors:** Wei-Qing Li, Jia-Yu Li, Yi-Fei Zhang, Wen-Qi Luo, Yi Dou, Song Yu

**Affiliations:** 1College of Agriculture, Heilongjiang Bayi Agricultural University, Daqing 163319, China; lwq_ljy@163.com (W.-Q.L.); byndljy@163.com (J.-Y.L.); byndlwq@163.com (W.-Q.L.); bynddy@163.com (Y.D.); yusong@byau.edu.cn (S.Y.); 2Key Laboratory of Low-Carbon Green Agriculture in Northeastern China, Ministry of Agriculture and Rural Affairs, Daqing 163319, China

**Keywords:** seed germination, DMTU, H_2_O_2_, maize, gene expression, radicle elongation

## Abstract

Reactive oxygen species (ROS) are an important part of adaptation to biotic and abiotic stresses and regulate seed germination through positive or negative signaling. Seed adaptation to abiotic stress may be mediated by hydrogen peroxide (H_2_O_2_). The effects of the ROS scavenger N,N′-dimethylthiourea (DMTU) on maize seed germination through endogenous H_2_O_2_ regulation is unclear. In this study, we investigated the effects of different doses of DMTU on seed endogenous H_2_O_2_ and radicle development parameters using two maize varieties (ZD958 and DMY1). The inhibitory effect of DMTU on the germination rate and radicle growth was dose-dependent. The inhibitory effect of DMTU on radicle growth ceased after transferring maize seeds from DMTU to a water medium. Histochemical analyses showed that DMTU eliminated stable H_2_O_2_ accumulation in the radicle sheaths and radicles. The activity of antioxidant enzyme and the expression of antioxidant enzyme-related genes (*ZmAPX2* and *ZmCAT2*) were reduced in maize seeds cultured with DMTU compared with normal culture conditions (0 mmol·dm^−3^ DMTU). We suggest the use of 200 mmol·dm^−3^ DMTU as an H_2_O_2_ scavenger to study the ROS equilibrium mechanisms during the germination of maize seeds, assisting in the future with the efficient development of plant growth regulators to enhance the seed germination performance of test maize varieties under abiotic stress.

## 1. Introduction

Germination is the first step in plant development and is a complex physiological and biochemical process [1,2,3,4,5]. Seed germination plays a critical role in seedling establishment and ensures that mature plants grow to their full potential, which is biologically, ecologically, and economically important. During germination, metabolic activity is reactivated, cellular respiration resumes, mitochondrial biogenesis is reactivated, DNA repair occurs, stored mRNAs are translated and/or degraded, new mRNAs are synthesized, and reserves are mobilized [6,7,8].

Reactive oxygen species (ROS), in addition to being crucial growth and development regulators for plants, are essential signaling mediators for plants to adapt to biotic and abiotic challenges [9,10,11,12,13]. Researchers discovered that ROS are crucial for seed cell wall thinning, endosperm softening, mobilization of seed storage components, disease resistance, and programmed cell death [14]. Hydrogen peroxide (H_2_O_2_), a kind of ROS, is regarded as the most important molecule due to its dual roles in stress sensing and plant tolerance to abiotic stressors [13]. Seed germination can be positively or negatively regulated by this signaling molecule [9]. Studies have revealed that H_2_O_2_ can induce memory imprinting in citrus seeds to stop harmful stress damage and works as a signal molecule to start germination [15,16,17]. Furthermore, because of its easy penetration into the cell for conversion into other free radicals, H_2_O_2_ is considered a signaling hub that regulates seed dormancy and germination and is therefore widely used as an oxide in studies of intracellular oxidative and antioxidant homeostasis. H_2_O_2_ also regulates many intracellular signaling pathways and redox-sensitive genes as a secondary messenger. In *Jatropha curcas* seeds, H_2_O_2_ is actively used to modulate the expression of antioxidants, redox balance genes, stress perception genes, and signal transduction genes during germination and initiation [18]. Therefore, it is vital for seed germination to maintain a balance between oxidative communication and oxidative damage by the precise regulation of H_2_O_2_ accumulation through cellular inhibition mechanisms [19,20].

Seeds germinate by imbibing water, and the emergence of embryonic axis and peri-embryonic components marks the completion of germination. During seed germination, the endosperm cap weakens, and the radicle lengthens, indicating the seed is transitioning from a stationary to a metabolically active state [18]. Therefore, to germinate, the endosperm cap must weaken, and the radicles must lengthen. These processes require the loosening of cell walls through enzymatic (such as endo-1,4-mannanase, pectin methyl esterase, and cellulase) and non-enzymatic (such as ROS, including •O_2_^−^, H_2_O_2_, and •OH) mechanisms [8,19,20,21]. In a previous study, H_2_O_2_ content and peroxidase activity increased rapidly in lettuce seed caps during germination; however, endosperm cap strength considerably decreased, suggesting that H_2_O_2_ production is closely related to endosperm weakening [22]. Histochemical staining revealed that H_2_O_2_ was dominant in rice embryos, endosperm, radicle sheaths, and radicles after germination. This tissue-specific accumulation of H_2_O_2_ can be inhibited by exogenous addition of diphenyleneiodonium chloride (DPI, an ROS generation inhibitor).

Maize is a major crop grown in most soil types and climates [23,24,25]. Maize crops provide food for humans and serve as an important source of industrial products, biofuels, and animal feed [26]. The global population is continuously increasing, and people’s living standards are gradually improving, increasing the demand for maize products. Therefore, ensuring sustained and stable maize production is crucial. The plant’s life cycle seed germination stage is the most vulnerable to environmental influences. In imbibition and germination, seeds are highly susceptible to destructive, abiotic, and biotic stresses, leading to excessive ROS accumulation; however, high or low ROS levels do not promote seed germination [27]. Furthermore, the level of ROS affected by environmental conditions determines the balance between required signaling events and oxidative damage [9,28,29]. Among ROS detoxification activities, catalase (CAT) and superoxide dismutase (SOD) act as front lines of antioxidant defenses, scavenging H_2_O_2_ and •O_2_^−^, respectively, and throughout plant development, they can be found in most tissues. Researchers have found that salt stress reduces radical length and plant height, as well as increasing H_2_O_2_ levels in maize seedlings [30]. In maize plants, three CAT isozymes and nine SOD encoding genes are regulated by genetic, developmental, and environmental factors [31,32]. In different tissues and developmental stages, they are regulated by stress (such as osmotic stress) and adversity-related growth regulators—such as abscisic acid (ABA)—[33,34,35,36]. The production of endogenous ROS can be induced by ABA, resulting in the activation of *Sod4*. *Sod4A* expression is only influenced by ABA in young maize leaves, and ABA is indirect in its effects [34]. However, the relationship between maize seed germination and the dynamic balance of ROS in different seed tissues is not well understood.

N,N′-dimethylthiourea (DMTU) is a ROS scavenger, including hydrogen peroxide (H_2_O_2_), hydroxyl radicals (•OH), etc., that mitigates ROS damage by reducing excess ROS accumulation in plants [37]. ROS production and microglial cell activation are obviously inhibited by pretreatment with DMTU [38].Notably, DMTU is considered the least toxic of alkylthioureas and is widely used in in vivo studies because of its low toxicity to biological systems [39,40]. In plant systems, DMTU was often used in previous studies to determine ROS homeostasis mechanisms during plant growth and development in adversity [41,42,43]. Furthermore, DMTU treatment can alleviate the effects of exogenous spermidine (Spd) on maize seedling growth to some extent [44]. The application of DMTU reversed the horizontal bending induced by exogenous H_2_O_2_ in the radical systems of grass pea (*Lathyrus sativus* L.) [45]. In *Medicago truncatula*, DMTU pretreatment prevented BR-induced increases in H_2_O_2_ and NO levels [46]. Furthermore, the expression and activity of the gene encoding NADPH oxidase was reduced by pretreatment with DMTU in grape seedlings under cold stress induced by BR [47]. However, few studies on the role of DMTUs in regulating endogenous H_2_O_2_ during maize seed germination have been conducted. There are also few studies on the role of DMTU in regulating endogenous H_2_O_2_ during maize seed germination. In northeastern China, especially in high-latitude cold regions, maize, as the largest food crop in the sown area, is produced by direct sowing technology, and the unfavorable environmental conditions, such as low temperatures, drought, and salinity, which often lead to a slowing of the germination process and a decrease in the emergence rate of maize in the spring, have become the main limiting factors hindering the construction of high-yielding maize populations in the region; the global climate change in recent years has further aggravated the adverse impacts of the abiotic stresses on the germination and establishment of maize seed and the sustained and high quality development of the maize industry in the region [48,49,50,51]. Therefore, in this study, two main maize varieties, Zhengdan 958 (ZD958) and Demeiya 1 (DMY1), which are widely used in the high-latitude cold region of northeast China, were selected to better understand the relationship between the signaling molecule H_2_O_2_ and the development of embryonic roots and seed germination characteristics of the above two maize varieties, investigating the effects of different concentrations of DMTU on the endogenous H_2_O_2_ and embryonic root growth characteristics of maize seeds. This study will provide a theoretical basis and technical reference for future research on seed germination characteristics and the intrinsic ROS balance mechanism of exogenous plant growth regulators on the above two maize varieties using DMTU as an H_2_O_2_ scavenger. The aim is to innovate efficiently and develop maize seed antistress control technologies, to improve the resistance of maize seeds to abiotic stresses after sowing in springtime in northeastern China, and to promote a steady increase in regional maize yields.

## 2. Results

### 2.1. DMTU Inhibits Maize Seed Growth by Inhibiting Germination and Radicle Extension

Research on maize seed germination and radicle elongation was conducted with sterile water (control) or DMTU at four dosages to examine the ROS scavenger (DMTU) effects. Under sterile water incubation conditions, the germination rates were 1.5%, 12%, 34%, 61% for ZD958 seeds, and 22%, 46%, 73.5%, 95.5% for DMY1 seeds at 24, 30, 36, and 42 h, respectively (Figure 1A,B).

Maize seeds treated with different DMTU concentrations showed significantly (*p* < 0.05) different germination rates at 36–42 h compared with control culture treatments, and germination rates declined under DMTU added conditions. With increasing DMTU concentration, the inhibitory effect on germination gradually increased. The germination rates of ZD958 and DMY1 under the addition of 0, 50, 100, 200, and 300 mmol·dm^−3^ DMTU were 61%, 53.5%; 47%, 35.5%; 12.5% 95.5%; 78%, 72%; 61%, 47.5%, respectively, up to 42 h of incubation. After 48 h, the germination rate difference between the DMTU-treated maize seeds and the control treatment was lessened, but the level of radicle growth inhibition remained elevated as the DMTU concentration rose (Figure 1C,D). To further explore the regulatory effects of DMTU on maize seed germination and radicle elongation, a 200 mmol·dm^−3^ DMTU treatment was chosen.

### 2.2. DMTU Reduces the Viability of Maize Seed Embryos and Radicles

Under sterile water control or 200 mmol·dm^−3^ DMTU culture conditions, we stained and photographed complete and half maize seeds with 2,3,5-triphenyltetrazolium chloride (TTC) to study how ROS scavengers (DMTU) affect seed viability (Figure 2). The staining results showed that ZD958 (Figure 2A) and DMY1 (Figure 2B) have similar staining results across tissues or parts. Under sterile water culture conditions, the embryo, radicle, and radicle sheath of the intact seeds of both varieties were stained; there was no staining of the starchy endosperm or dextrin layer, indicating the embryo and radicle were more involved in maize seed germination. When seeds were cultured with the addition of 200 mmol·dm^−3^ DMTU, embryos, radicles, and radicle sheaths were weakly stained at 12 h and 48 h compared to cultures in sterile water control environments, whereas the dextrin layer and the starchy endosperm in cross sections of both varieties did not appear stained. Furthermore, this study showed that DMTU was specifically responsible for reducing embryo viability, radicle sheath viability, and radicle viability during maize seed germination but was not responsible for affecting starchy endosperm and aleurone viability.

### 2.3. DMTU Inhibits H_2_O_2_ Production and Accumulation in the Maize Seed Embryo, the Radicle, and the Radicle Sheath

The production and the accumulation of H_2_O_2_ during maize seed germination were observed using 3,3-diaminobenzidine (DAB) staining. After the growth of seeds in sterile water, both ZD958 and DMY1 seed embryos gradually became more intensely stained as germination progressed (Figure 3). However, when maize seeds were incubated at 200 mmol·dm^−3^ DMTU for 48 h, compared to the sterile water control condition, embryo, radicle, and radicle sheath staining intensity was reduced. Notably, 200 mmol·dm^−3^ DMTU treatment did not inhibit or inhibit to a lesser extent the staining intensity of the seed aleurone layer compared to seeds cultured in sterile water, and ZD958 and DMY1 seeds exhibited the same pattern (Figure 3A,C). After 48 h of incubation in sterile water, the H_2_O_2_ levels in the embryos of ZD958 and DMY1 seeds (Figure 3B,D) reached their peak; the H_2_O_2_ concentration in maize seed embryos cultivated under 200 mmol·dm^−3^ DMTU conditions was considerably lower (*p* < 0.05).

### 2.4. DMTU Scavenges H_2_O_2_ in Maize Seed Embryos and Inhibits Radicle Elongation and Growth

We investigated whether DMTU could inhibit the elongation and growth of radicles derived from maize seeds and this is shown in Figure 4A,C. The radicles of the ZD958 and DMY1 seeds elongated rapidly after 72, 120, and 144 h in sterile water, indicating strong radicle growth. After 72, 120, and 144 h, the seed radicles of ZD958 and DMY1 rapidly elongated in sterile water, suggesting a strong growth potential. Incubation in DMTU for 72 h, followed by transfer to sterile water, resulted in radicle growth potential in ZD958 and DMY1 cells, suggesting that their growth capacity was restored.

Therefore, for incubation for 72, 120, and 144 h, we measured the length of the ZD958 and DMY1 radicles simultaneously. According to Figure 4B,C,E,F, when ZD958 and DMY1 were cultured in sterile water, the length of the radicle gradually increased with time, reaching its maximum value after 144 h of incubation, which was 14.45 cm and 14.07 cm, respectively; furthermore, the H_2_O_2_ content remained high. However, when maize seeds were cultured under DMTU conditions of 0 mmol·dm^−3^ DMTU, radicle elongation and growth were significantly inhibited (*p* < 0.05) in both ZD958 and DMY1. At 144 h of incubation, the radicle lengths of ZD958 and DMY1 were only 2.53 cm and 2.08 cm, and the H_2_O_2_ content was also significantly lower (*p* < 0.05) than in sterile water culture conditions. However, when the maize seeds incubated under DMTU conditions for 72 h were transferred to sterile water for further cultivation, the radicle gradually resumed to elongate and grow, and the radicle length of ZD958 and DMY1 reached 9.92 cm and 9.68 cm at 144 h, respectively. When the radicle was transferred from DMTU to sterile water for cultivation, the H_2_O_2_ content in the radicle increased rapidly, which was consistent with the morphological changes. Thus, results show that DMTU can inhibit radicle elongation and growth during maize seed germination by specifically removing H_2_O_2_ from the seeds.

### 2.5. Inhibition of Antioxidant Enzymes during Maize Seed Germination by DMTU

In ZD958 and DMY1 maize seeds cultured in sterile water, the activities of antioxidant enzymes, such as peroxidase (POD), catalase (CAT), and ascorbate peroxidase (APX), increased as the incubation time was extended. After 48 h of incubation, antioxidant enzyme activity peaked (Figure 5). When incubated with 200 mmol·dm^−3^ DMTU, both varieties had significantly lower POD, CAT, and APX enzyme activities (*p* < 0.05) than when incubated with sterile water culture. Furthermore, both varieties showed a continuous decline in SOD activity with increasing incubation time. Compared with sterile water cultivation, the SOD enzyme activity of each variety did not show remarkable changes after 24 and 48 h of incubation under 200 mmol·dm^−3^ DMTU culture conditions.

### 2.6. Expression of ZmSOD4, ZmCAT2, and ZmAPX2 during Maize Seed Germination

As shown in Figure 6, we measured the expression levels of antioxidant enzyme-related genes *ZmSOD4*, *ZmCAT2*, and *ZmAPX2*. As seed cultivation time increased under sterile water culture conditions, *ZmSOD4* expression decreased and then increased. However, *ZmSOD4* expression in ZD958 after 48 h of DMTU incubation was three times higher than that in the control treatment (Figure 6B), whereas that in DMY1 after 24 h of control treatment was higher (*p* < 0.05; 89 times) than that in DMTU incubation (Figure 6E). Compared to *ZmSOD4*, the expression of *ZmCAT2* and *ZmAPX2* was maintained at higher levels in ZD958 after 24 and 48 h of incubation in sterile water (Figure 6A,C), with the expression of *ZmCAT2* significantly (*p* < 0.05) higher than DMTU treatment by approximately 3-fold and 27-fold after 24 and 48 h of incubation, respectively. *ZmCAT2* and *ZmAPX2* expression patterns in DMY1 seeds had the same pattern as ZD958 seeds (Figure 6D,F), and both varieties of seeds showed the highest expression levels of *ZmCAT2* and *ZmAPX2* after 48 h of incubation in sterile water.

## 3. Discussion

In modern times, agricultural producers commonly use direct seeding technology for maize cultivation [52,53]. Because seedling growth and germination depend highly on the conditions in the field, especially low temperatures, droughts, and other adverse conditions, maize seed germination, seedling establishment, and yield will suffer severe adverse effects. During seed germination, ROS are continuously produced and scavenged to reach a dynamically balanced level that affects seed germination. According to earlier research, the exogenous administration of H_2_O_2_ can facilitate the germination of pea (*Pisum sativum* L.) seeds and seedling growth [54]. Furthermore, the exogenous addition of H_2_O_2_ improves the germination performance of dormant seeds because non-dormant seeds produce more H_2_O_2_ molecules than dormant seeds during imbibition [9]; it involves endogenous H_2_O_2_ production and the impact of significant H_2_O_2_ accumulation on seed germination potential. However, little is understood about how an H_2_O_2_ scavenger, DMTU, affects the germination of maize seeds. Therefore, in this study, we investigated thoroughly the ROS homeostasis mechanism of DMTU by adding various doses of DMTU during the germination of maize seeds and the subsequent seedling growth.

Through the maize seed germination experiments, we initially found that 50–300 mmol·dm^−3^ DMTU treatments all delayed maize seed germination to varying degrees (Figure 1A,B), and as the concentration of DMTU increased, the rate at which maize seeds germinated dropped. According to DMTU, maize seeds are removed specifically from endogenous H_2_O_2_ during germination, resulting in lower levels of H_2_O_2_ involved in germination, causing less pronounced inhibition of germination. Increased concentrations of DMTU led to a lower endogenous H_2_O_2_ content, resulting in stronger inhibition. In these experiments, DMTU inhibited maize seed germination and growth, particularly radicle elongation; however, the seeds resumed growth when removed from the culture environment. According to the theory of “germination oxidation window”, germination can be hindered by low or high ROS levels [9]; we speculated that the slowing down of the seed germination process after the addition of DMTU in the culture environment may be related to the altered redox balance in different tissue parts of the maize seeds.

The beginning of seed germination is thought to be the uptake of water, whereas the completion of germination is thought to be the emergence of the axis of the embryo and peri-embryonic structures [55]. According to previous studies, the criteria for completing germination are different for dicotyledons and monocotyledons, such as lettuce (*Lactuca sativa* L.) [19], watercress (*Lepidium sativum* L.) [56], and tomato (*Solanum lycopersicum* L.) [57], which have a high mechanical strength in the endosperm that prevents germination from being completed. Thus, for seed germination, micropylar endosperm softening and radicle elongation are crucial. However, for monocotyledons such as rice (*Oryza sativa* ssp.), barley (*Hordeum vulgare* L.), and wheat (*Triticum turgidum* L.), the endosperm provides nutrition for germination, the emergence of peri-embryonic structures, and the growth of young vegetative bodies [58]. In maize, a monocotyledonous plant, during seed germination, the radicle sheath first appears and then emerges from the radicle sheath when the seed has fully germinated. Recent research studies strongly believe that the radicle sheath plays a part in inhibiting radical emergence and expansion during seed germination; during the germination of rice seeds, the radicle sheath may serve as a barrier to radicle protrusion and successful germination [59], which appears similar to the role of the bead hole endosperm in dicotyledons. In this study, during maize seed germination, the radicle sheath gradually disappeared after 24 h of seed imbibition (particularly after the emergence of the radicle from the radicle sheath), similar to that reported in other monocotyledons, such as rice and barley. Seed vigor is the potential to produce vigorous seedlings; as an important agronomic trait with direct relevance to production, it is commonly used to predict the germinability of various agricultural seeds [60]. A previous study aimed to investigate the effect of ROS on seed viability in SO_2_-promoted maize seed germination; diphenyleneiodinium (DPI), an inhibitor of ROS production, was applied to examine the viability of germinated seeds using TTC staining. The embryos of the seeds co-treated with DPI and SO_2_ showed considerably lower TTC staining intensities than those of the SO_2_ or control treatment. This finding suggests that DPI inhibits the generation of ROS and reduces the viability of the embryos of maize seeds in the process of seed germination [61]. In this study, it was found through TTC staining that the viability of maize seed embryos gradually increased with increasing incubation time, and the viability of the embryos of germinating maize seeds incubated in water was higher than that of the embryos of DMTU-treated germinating maize seeds (Figure 2A,B). This finding suggests that DMTU leads to a loss of seed viability and thus delays germination. Furthermore, with an increase in incubation time, the H_2_O_2_ content in maize seed embryos gradually increased, and the level of H_2_O_2_ accumulation in the radicle sheath and radicle was higher than in germinated maize seeds treated with DMTU (Figure 3B,D and Figure 4C,F). These results suggest that non-enzymatic mechanisms, especially moderate accumulation of H_2_O_2_, are necessary for the successful completion of maize seed germination. Furthermore, studies on dicotyledonous lettuce seeds have also revealed that H_2_O_2_ has a role in the cell wall loosening of the radicle sheath and the radicle of maize seeds, facilitating the germination process and seedling growth [15].

Under abiotic stressor conditions, H_2_O_2_ is regarded as a good regulator of antioxidant defense mechanisms (enzymatic and non-enzymatic) [16,62]. H_2_O_2_ stimulates the antioxidant defense mechanism in seeds, which is crucial for seed vigor and survival. Superoxide anions (•O_2_^−^) can be transformed into H_2_O_2_ by SOD, which can then be scavenged by POD in the extracellular region. CAT converts H_2_O_2_ to H_2_O and O_2_, whereas APX uses ascorbic acid as a donor to remove H_2_O_2_ [63]. This study used DMTU to scavenge H_2_O_2_ signaling, which adversely affected H_2_O_2_ signaling (Figure 3B,D). The addition of exogenous DMTU accelerated the reduction in endogenous H_2_O_2_ content and inhibited radicle elongation and growth (Figure 1C,D). A previous study in tomato (*Solanum lycopersicum* L.) plants reported that DMTU reduced the cold-induced overaccumulation of H_2_O_2_ in tomato leaves and also inhibited the cold-induced increase in the activities of antioxidant enzyme activities such as SOD, CAT, and APX [64]. We investigated H_2_O_2_-related antioxidant enzymes and the antioxidant enzyme genes *ZmSOD4*, *ZmAPX2*, and *ZmCAT2* during maize seed germination to verify the role of H_2_O_2_ as a signaling molecule. *ZmSOD4*, *ZmAPX2*, and *ZmCAT2* are involved in the stress response in maize [65], and in maize seedlings cultured in sterile water; the expression of *ZmAPX2* and *ZmCAT2* increases with time. However, this expression trend was affected by DMTU treatment. The expression of *ZmAPX2* and *ZmCAT2* decreased at different time points after the DMTU treatment. In addition, DMTU decreased the POD, CAT, and APX antioxidant enzyme activity, which may be attributed to DMTU specifically scavenging endogenous H_2_O_2_ during the germination of maize seeds, leading to a decrease in downstream antioxidant enzyme activity. The expression profile of *ZmSOD4* appeared to be unaffected by the addition of DMTU, which may be because stable •O_2_^−^ is produced during seed germination under normal conditions and the need for a certain amount of SOD to convert •O_2_^−^ into H_2_O_2_ to promote the germination process. As an H_2_O_2_ scavenger, DMTU interferes only with the metabolic processes and pathways downstream of the H_2_O_2_ signal. In summary, under the conditions used in the present study, exogenous addition of 200 mmol·dm^−3^ DMTU retarded maize seed germination and radicle elongation by downregulating endogenous H_2_O_2_ signaling in the embryos of the test maize seeds and negatively regulating the gene expression and activities of downstream related antioxidant enzymes (Figure 7).

## 4. Materials and Methods

### 4.1. Non-Plant Materials

All the chemical reagents used in this study were of the analytical grade. N,N′-dimethylthiourea (DMTU), 3,3-diaminobenzidine (DAB), 2,3,5-triphenyl-2H-tetrazolium chloride (TTC) were obtained from Sigma-Aldrich (St. Louis, MO, USA). High temperatures (120 °C) were used to sterilize all the water used during the experiment.

### 4.2. Seed Germination

The maize seeds tested, Zhengdan 958 (*Zea mays* L. cv. ZD958, dent maize) and Demeiya 1 (*Zea mays* L. cv. DMY1, flint maize), were used as test maize varieties, and uniform and consistent seeds for each variety (40 and 27 g per 100 seeds for ZD958 and DMY1, respectively) were selected. Before experimentation, all seeds underwent 1% NaClO sterilization for 20 min, 75% (*v*/*v*) ethanol surface sterilization for 60 s, and six sterile water rinses. Subsequently, the seeds, after surface sterilization, were placed in a paper bed germination box (length × width × height: 14 cm × 14 cm × 6 cm) with 10 mL of sterile water containing 0, 50, 100, 200, and 300 mmol·dm^−3^ DMTU, and incubated in an artificial climate chamber (DRX-1000, Ningbo Dongnan Instrument Co., Ltd., Ningbo, China) in the dark (temperature 25 ± 0.5 °C, relative humidity 50 ± 5). Four replicates were used for each treatment, with each germination box serving as a single replicate. The dynamics of seed germination (radicles breaking through the seed coat of seeds) treated with different concentrations of DMTU were observed, and each 6 h span was calculated to determine the cumulative germination rate within 48 h. A stereoscopic zoom microscope system (Ste REO Discovery V8, Carl Zeiss AG, Oberkochen, Germany) was used at different time points to observe each treatment condition.

### 4.3. TTC Staining

TTC is a lipid-soluble photosensitive complex commonly used to test the viability of plant and animal cells or tissues. After treatment with sterile water control (0 mmol·dm^−3^ DMTU) or different concentrations of DMTU, 10 intact seeds and 10 embryonic half seeds were taken and immersed in a solution containing 0.2% TTC for 20 min at 30 °C, protected from light, then repeatedly rinsed with sterile water until the seed surface was free of stain, and finally photographed using a stereo-zoom microscope system (Ste REO Discovery V8, Carl Zeiss AG).

### 4.4. Histochemical Location of H_2_O_2_

Using a modified version of the 3,3-diaminobenzidine (DAB) method proposed by Xia et al. [66], H_2_O_2_ was detected. After treatment with sterile water control (0 mmol·dm^−3^ DMTU) or different concentrations of DMTU, 10 complete seeds and 10 embryonic half seeds were taken and soaked for 10 h at 25 °C using 1% DAB solution (pH 3.8). The seeds were rinsed repeatedly and photographed using a stereo-zoom microscope (Ste REO Discovery V8; Carl Zeiss AG).

### 4.5. H_2_O_2_ Content Measurement

According to Zhang et al. [22], H_2_O_2_ content was determined using the colorimetric method for maize with different treatments (control and DMTU-treated). H_2_O_2_ was extracted from germinating maize seeds (approximately 0.5 g) by homogenizing them in 3 cm^3^ of chilled acetone. To homogenize the samples, a tissue homogenizer (Tissuelyser-24, Jingxin Industrial Development Co. Ltd., Shanghai, China) was used at 4 °C and immediately centrifuged for 15 min at 12,000× *g*. The peroxide-titanium complex was precipitated using aqueous NH_3_ (25%), 0.2 cm^3^ concentrated HCl, and 1 cm^3^ of the supernatant solution. Precipitates were dissolved in 2 mmol·dm^−3^ H_2_SO_4_ and measured at 415 nm with a Specord Plus 210 (Analytik Jena AG, Jena, Germany) apparatus. This method was used to prepare the standard response curve for H_2_O_2_.

### 4.6. Antioxidant Enzyme Activity Determination

Seed maize embryos (approximately 3 g) were cultured under control or DMTU-treated conditions and ground into powder using liquid nitrogen. Following that, the powder was mixed at ice-cold temperatures in 30 cm^3^ of an obtaining buffer, which contained the following: 0.1 mol·dm^−3^ Tris–HCl (pH 7.5), 0.23 mol·dm^−3^ sucrose, 5% polyvinylpyrrolidone, 1 mmol·dm^−3^ EDTA, 10 mmol·dm^−3^ KCl, 10 mmol·dm^−3^ MgCl_2_, and 2.5 mmol·dm^−3^ ascorbic acid (added just before using the buffer). The supernatants underwent centrifugation and homogenization for enzyme assays (12,000× *g* for 20 min). All enzyme activities were determined by spectrophotometry using Specord Plus 210 (Analytik Jena AG).

Different quantities of enzyme extract were combined with the reaction buffer in 50 mmol·dm^−3^ phosphate buffer (pH 7.8) for the determination of SOD activity (EC 1.15.1.1), which contained 13 mmol·dm^−3^ methionine, 75 μmol·dm^−3^ nitro blue tetrazolium, 100 μmol·dm^−3^ EDTA, and 2 μmol·dm^−3^ riboflavin. The mixture was held up to daylight for 15 min. Then, the absorbance measured at 560 nm increased in the presence of formazan. In this study, a unit of SOD activity was defined as the amount of the enzyme that prevented 50% of nitro blue tetrazolium from being photo reduced [67].

POD activity (EC 1.11.1.7) was assessed mostly in accordance with Tan et al. [68] with a few adjustments. An enzyme extract of 40 mm^3^ was combined with 0.2 mol·dm^−3^ phosphate buffer (pH 6.0), 30% H_2_O_2_, and 50 mmol·dm^−3^ guaiacol and then measured at 470 nm for absorbance.

Catalase (CAT; EC 1.11.1.6) activity was estimated by combining 150 mmol·dm^−3^ of potassium phosphate buffer (pH 7.0), 15 mmol·dm^−3^ H_2_O_2_, and 50 mm^3^ of enzyme extract. Then, 15 mmol·dm^−3^ H_2_O_2_ was used to initiate the reaction. To measure CAT activity, H_2_O_2_ consumption was measured at 240 nm for 3 min [69].

In this study, the method used by Nakano and Asada [70] was used to assess the activity of APX (EC 1.11.1.1). Briefly, 50 mmol·dm^−3^ of phosphate buffer (pH 6.0), 0.1 mmol·dm^−3^ of EDTA-Na_2_, 5 mmol·dm^−3^ of AsA, and 20 mmol·dm^−3^ of H_2_O_2_ solution were combined with 0.1 cm^3^ of enzyme extract. After 40 s, the absorbance at 290 nm was calculated.

### 4.7. Gene Expression Analysis

Total RNA from embryos of maize seed samples treated with sterile water control (0 mmol·dm^−3^ DMTU) or 200 mmol·dm^−3^ DMTU at 0, 24, and 48 h was also extracted using TRIzol^®^ Reagents (Invitrogen, Carlsbad, CA, USA). Qualified RNA was used for reverse transcription using the ReverTra Ace^TM^ qPCR RT Master Mix with a gDNA Remover kit (TOYOBO Co., Osaka, Japan) after detection using a NanoDrop spectrophotometer (Thermo Fisher Scientific, Waltham, MA, USA), and 1% agarose for RNA quality. Primer-BLAST “http://www.ncbi.nlm.nih.gov/tools/primer-blast/ (accessed on 8 December 2022)” was used to design primers based on the cDNA sequences of *ZmSOD4*, *ZmAPX2*, and *ZmCAT2* (Appendix A); *ZmActin* [65] was used (Appendix A) as an internal reference gene. Subsequently, quantitative real-time PCR (qRT-PCR) was performed on a CFX96 real-time fluorescence quantitative PCR detection system (Bio-Rad Laboratories Inc., Hercules, CA, USA) with SYBR^®^ Green PCR Master Mix (TOYOBO Co.), according to the manufacturer’s instructions. The following conditions were used for PCR cycling: 95 °C for 60 s, followed by 39 cycles of 95 °C for 15 s, and 55 °C for 30 s. Each treatment contained four biological replicate samples (each biological replicate involved three technical replicates). Gene expression levels were calculated using the 2^−ΔΔCt^ method [71].

### 4.8. Statistical Analysis 

For every experimental treatment, four replicates were conducted, and all plant phenotypes, physiological and biochemical indicator values were calculated using the mean ± standard deviation (SD) of four replicates. IBM SPSS Statistics for Windows, version 19.0 (IBM Corp., Armonk, NY, USA) was used for the analysis of variance. The new multiple range test by Duncan was used to compare significant differences between treatments; at *p* < 0.05, there were significant differences between the groups. To test the significance of differences in qRT-PCR data, we used Student’s *t*-test (* *p* < 0.05, ** *p* < 0.01).

## 5. Conclusions

According to this study, DMTU (200 mmol·dm^−3^) impeded seed germination and reduced germination rate of the tested maize varieties ZD958 and DMY1. Furthermore, the inhibitory effect of DMTU on the seed germination of the test maize varieties was dose-dependent, indicating that the inhibitory effect of DMTU was stronger at higher concentrations. However, this inhibition was reversible. Maize seeds produce moderate amounts of H_2_O_2_ when germinated under normal conditions. DMTU significantly reduced the endogenous H_2_O_2_ content, the expression of *ZmAPX2*, *ZmCAT2*, and the activities of POD, CAT, and APX antioxidant enzymes in ZD958 and DMY1 maize seeds, inhibiting the elongation and growth of the radicles during maize seed germination. This study confirmed that 200 mmol·dm^−3^ DMTU was actively involved in the regulation of downstream antioxidant enzyme activities of H_2_O_2_ scavenge through H_2_O_2_ signaling, thus affecting seed germination and radicle growth of ZD958 and DMY1, grown mainly in the high-latitude cold regions of northeast China. Furthermore, it provided an opportunity to analyze the effects of exogenous plant growth regulators on the seed germination characteristics of the test maize varieties and the intrinsic ROS equilibrium mechanism of the test maize varieties. It assisted in the development of the technology to regulate seed germination capacity and resistance of the test varieties, and to facilitate the steady improvement of regional maize yields, using DMTU at 200 mmol·dm^−3^ as a scavenger for H_2_O_2_ with future potential.

## Figures and Tables

**Figure 1 ijms-24-15557-f001:**
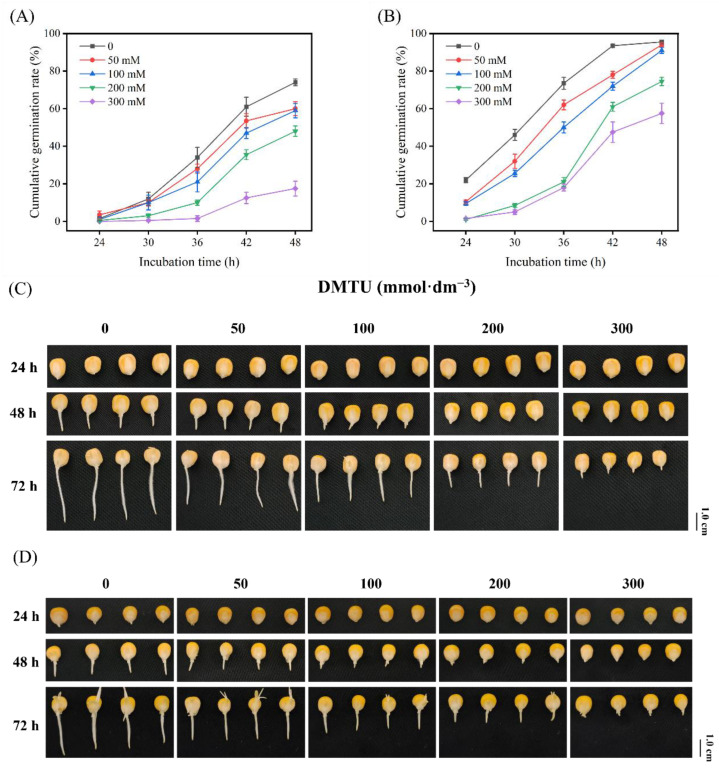
Germination rate and morphological characteristics of maize seeds at different time points under sterile water and ROS scavenger (DMTU) culture conditions. ZD958 (**A**) and DMY1 (**B**) seeds were cultured under sterile water (control) or different ROS scavenger (DMTU) concentrations to investigate seed germination rates at different time points. Germination was monitored every 6 h after 48 h. The cumulative germination rate was calculated based on these results. Data represent the mean of four replicates for each of the 50 seeds. Morphological characteristics of ZD958 (**C**) and DMY1 (**D**) seeds were observed and photographed under sterile water (control) or different ROS scavenger (DMTU) concentration culture conditions at 24, 48, and 72 h. DMTU—N,N′-dimethylthiourea. Scale bar, 1.0 cm.

**Figure 2 ijms-24-15557-f002:**
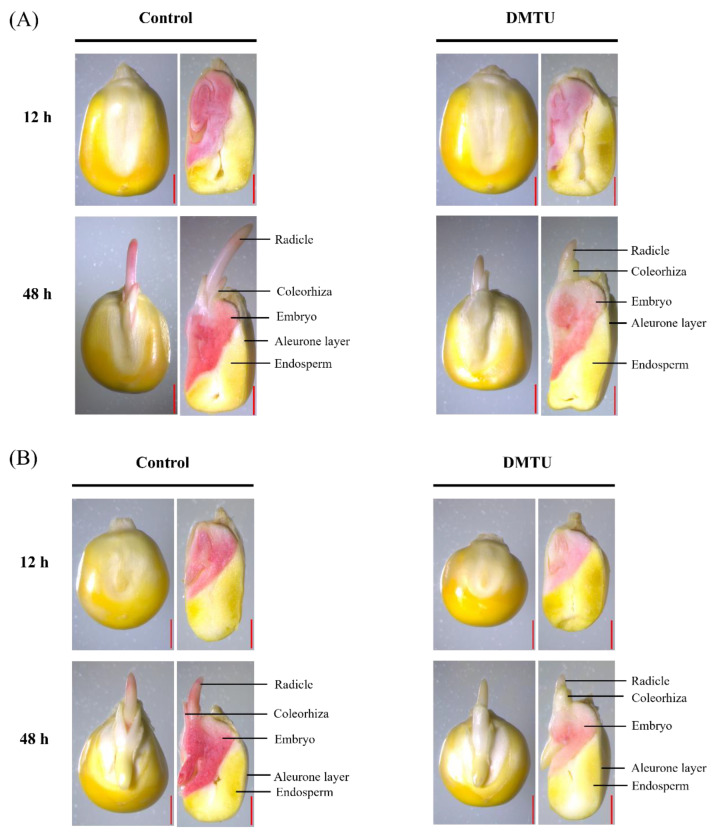
Different times of staining of maize seed with 2,3,5-triphenyltetrazolium chloride (TTC) in sterile water and ROS scavenger (DMTU). ZD958 (**A**) and DMY1 (**B**) were incubated in sterile water (control) or ROS scavenger (DMTU) for 12 or 48 h. We stained complete seeds or half of the seeds with 2,3,5-triphenyltetrazolium chloride (TTC) in both varieties. Dark red staining indicates high cell or tissue viability, whereas light pink staining indicates low cell or tissue viability. DMTU—N,N′-dimethylthiourea. Scale bar, 100 µm.

**Figure 3 ijms-24-15557-f003:**
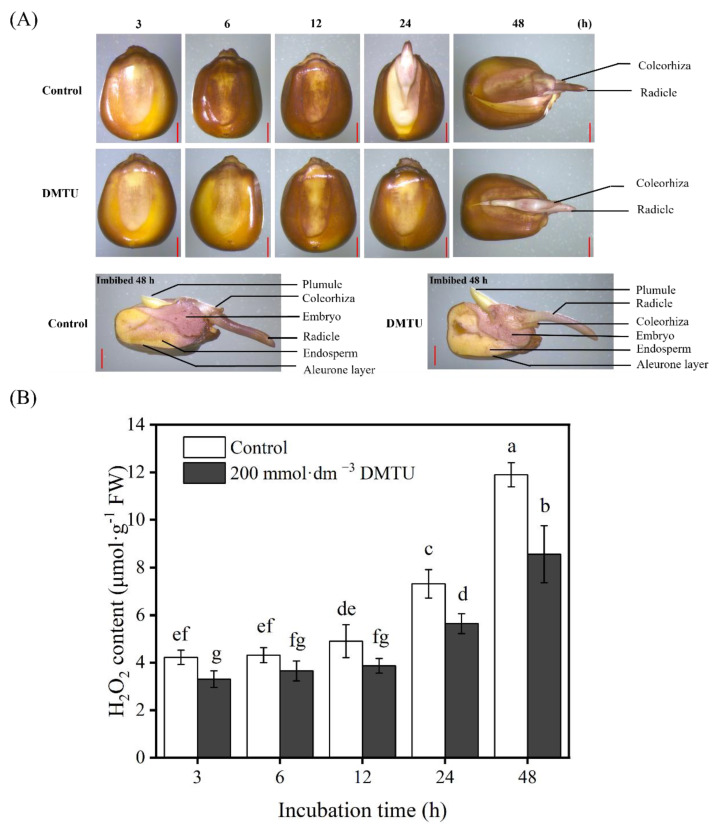
Maize seeds stained with histochemical staining and quantified using ROS scavenger DMTU culture. After incubation in sterile water (control) and ROS scavengers (DMTU) for 3, 6, 12, 24, and 48 h, the complete or half of the seeds of ZD958 (**A**) and DMY1 (**C**) were stained with 3,3-diaminobenzidine (DAB). Quantitative detection of H_2_O_2_ was performed in ZD958 (**B**) and DMY1 (**D**) seed embryos after incubation in sterile water (control) and ROS scavenger (DMTU) conditions for 3, 6, 12, 24, or 48 h. Data are the mean standard deviation of five embryos (approximately 0.1 g) from four replicates. Between treatments and incubation time points, there was significant variance in the means corresponding to different letters (*p* < 0.05, Duncan’s multiple range test). FW—fresh weight. DMTU—N,N-dimethylthiourea. Scale bar, 100 µm.

**Figure 4 ijms-24-15557-f004:**
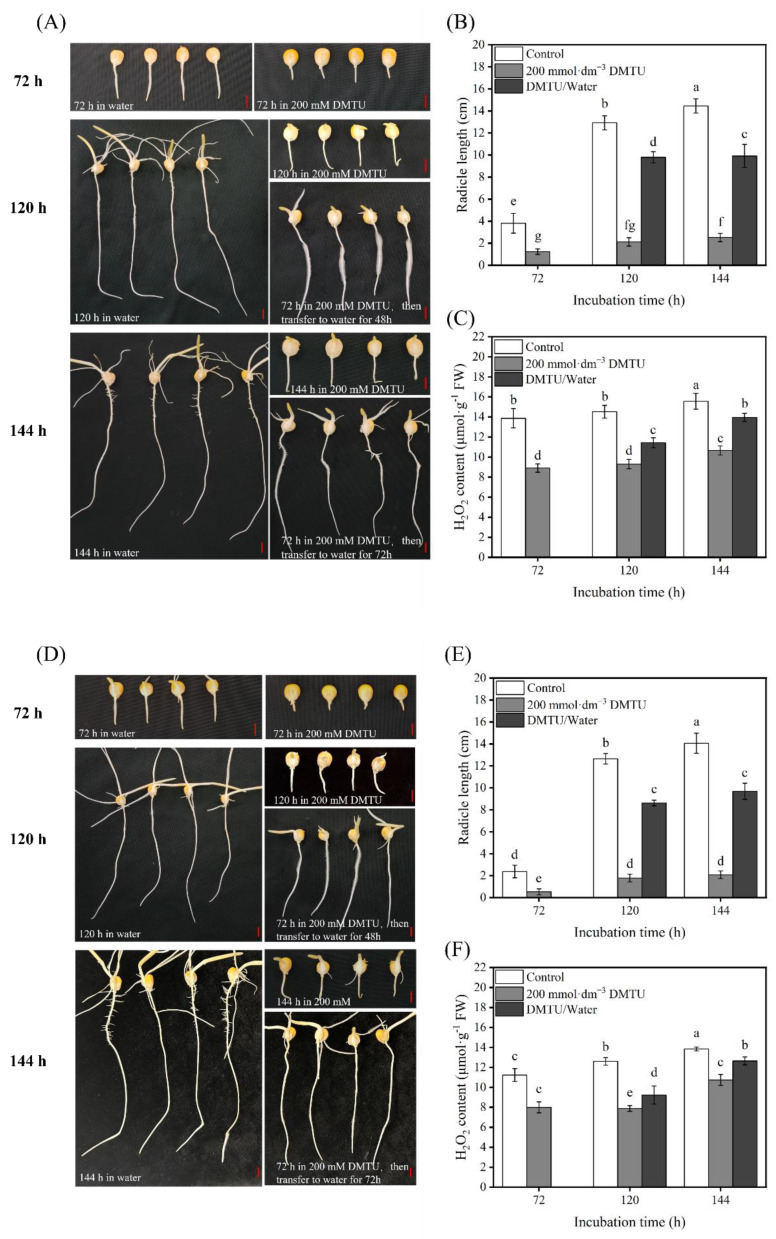
Morphological characteristics, radicle length, and H_2_O_2_ content of maize seeds cultured in sterile water and ROS scavenger (DMTU) for 72, 120, and 144 h. Morphological characteristics of ZD958 (**A**) and DMY1 (**D**) seeds were observed and photographed after incubation in sterile water (control) and ROS scavenger (DMTU) conditions for 72, 120, and 144 h. Radicle lengths of ZD958 (**B**) and DMY1 (**E**) seeds were determined after incubation in sterile water (control) and ROS scavengers (DMTU) for 72, 120, and 144 h, respectively. The H_2_O_2_ content in the radicles of ZD958 (**C**) and DMY1 (**F**) seeds was measured after incubation in sterile water (control) and ROS scavenger (DMTU) for 72, 120, and 144 h. In DMTU/water, maize seeds were incubated in ROS scavenger (DMTU) for 72 h, and then transferred to sterile water and incubated for another 48 and 72 h. Data are presented as the mean ± standard deviation of five embryos (about 0.1 g) from four replicates. Between treatments and incubation durations, there were significant differences in the mean of different letters (*p* < 0.05). FW—fresh weight. DMTU—N,N-dimethylthiourea. Scale bar, 100 µm.

**Figure 5 ijms-24-15557-f005:**
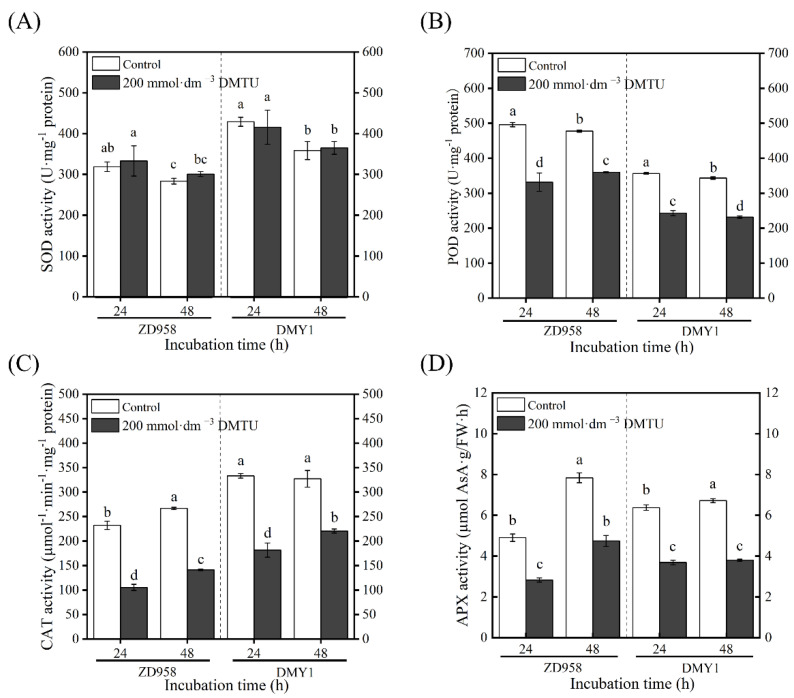
Antioxidant enzyme activity of maize seeds after 24 and 48 h of incubation with sterile water and ROS scavenger (DMTU). The ZD958 and DMY1 seed embryos incubated in sterile water (control) or ROS scavenger (DMTU) for 24 and 48 h demonstrated SOD (**A**), POD (**B**), CAT (**C**), and APX (**D**) activities. Data are the mean standard deviation of five embryos (about 0.1 g) from four replicates. Between treatments and incubation durations, there were significant differences in the mean of different letters (*p* < 0.05). FW—fresh weight. DMTU—N,N-dimethylthiourea; SOD—superoxide dismutase; POD—peroxidase; CAT—catalase; APX—ascorbate peroxidase.

**Figure 6 ijms-24-15557-f006:**
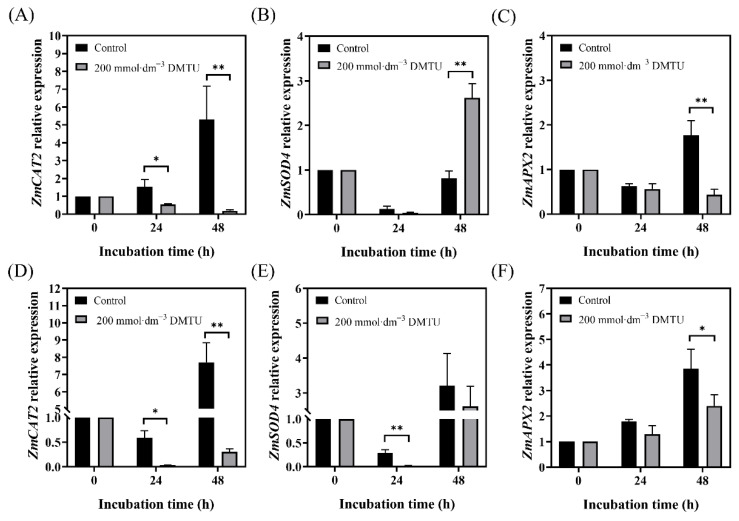
*ZmSOD4*, *ZmCAT2*, and *ZmAPX2* expression levels in seeds of maize after 24 and 48 h of ROS scavenger (DMTU) and sterile water incubation. Expression levels of *ZmCAT2* in ZD958 (**A**) and DMY1 (**D**) seed embryos were determined after incubation in sterile water (control) and ROS scavenger (DMTU) for 24 and 48 h, respectively. The expression levels of *ZmSOD4* in ZD958 (**B**) and DMY1 (**E**) seed embryos were determined after incubation in sterile water (control) and ROS scavenger (DMTU) for 24 and 48 h, respectively. Expression levels of *ZmAPX2* in ZD958 (**C**) and DMY1 (**F**) seed embryos were determined after incubation in sterile water (control) and ROS scavenger (DMTU) for 24 and 48 h, respectively. Data are presented as the mean ± standard deviation of five embryos (about 0.1 g) from four replicates. Using the Student’s *t*-test, we assessed whether there were significant differences between the 24 and 48 h qPCR results and the 0 h qPCR data (* *p* < 0.05; ** *p* < 0.01). DMTU—N,N-dimethylthiourea.

**Figure 7 ijms-24-15557-f007:**
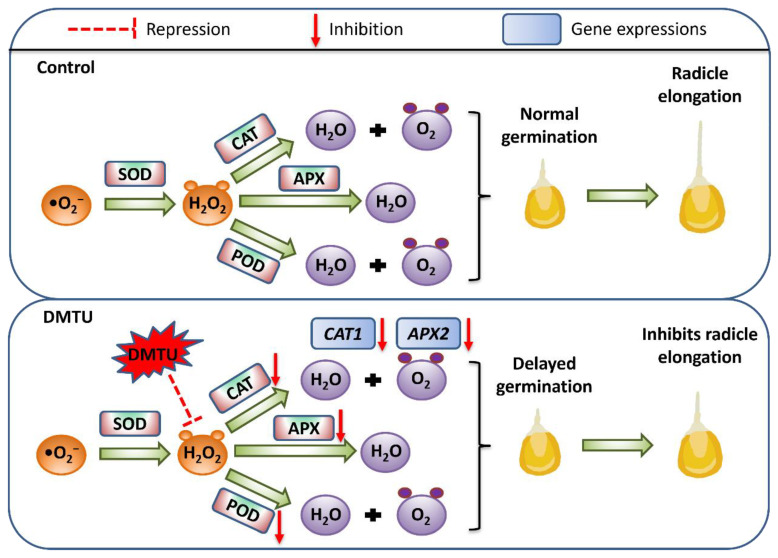
A model showing the endogenous scavenging of H_2_O_2_ by DMTU during maize seed germination using sterile water (control) and ROS scavenger (DMTU). H_2_O_2_—hydrogen peroxide; •O_2_^−^—superoxide anion; SOD—superoxide dismutase; POD—peroxidase; CAT—catalase; APX—ascorbate peroxidase.

## Data Availability

Not applicable.

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
