# Peer review of "Effect of Reactive Oxygen Scavenger N,N′-Dimethylthiourea (DMTU) on Seed Germination and Radicle Elongation of Maize"

_ijms, 2023, doi:10.3390/ijms242115557_

Round 1
Reviewer 1 Report
The MS could be acceptable for publication if the authors address the following points:
1) DMTU concentrations used in this study are much higher than in similar works. Could this be the primary reason for inhibition of seed germination? How do the authors explain their choice of DMTU concentrations?
2) Is DMTU cell toxicity (especially in the chosen concentration range) limited to ROS scavenging? DMTU also has the ability to form hydrogen bonds with other molecules, which may contribute to its cytotoxic activity.
3) The choice of maize cultivars should also be explained at the end of Introduction section.
4) lines 125-127: The germination rates of ZD958 at 300 mM DMTU was 95.5? Data in Figure 1A suggests it was ca.15%.
5) lines 193-195: what do the authors mean by "radicle growth potential"?
The text should be checked by a native English speaker. The meaning of some sentences is unclear.
Author Response
请参阅附件。

Reviewer 2 Report
Dear Authors,
The paper discusses the multi-step biological process of germination, which involves enzymes, non-enzymes, signal transduction, and gene expression mechanisms. Reactive oxygen species (ROS) play an important role in plant adaptation to biotic and abiotic stressors, as well as in regulating seed germination through positive or negative signaling.
Main objection: The research article presents an investigation primarily focusing on the utilization of seeds from what seems to be a two variety. Maize seeds, however, exhibit substantial diversity due to their classification into sub-species like ssp. indentata, indurata, saccharata, everta, amylacea, certina, tunicata, and amyleo-saccharata. Moreover, within these sub-species, there can be significant genetic variations among different varieties. This commentary seeks to highlight the importance of recognizing and addressing this genetic diversity in maize seeds, which appears to have been overlooked in the original work.One of the main concerns raised by the reviewer is the absence of information regarding the specific maize varieties used in the research. The use of seeds from only two variety may not accurately represent the broader genetic diversity within the maize species.
Other comments:
-based on studies of two variety the authors talk about the entire corn species in the title and conclusions which seems like too far of a generalization (see main objection),
- I lack a clear specification of the purpose of the research/research hypothesis and the reasons for undertaking this research, which would be compatible with the presented conclusions,
- in the conclusions the sentences "According to this study, DMTU (200 mmol·dm−3) impeded maize seed germination and decreased germination rate" and "Furthermore, the DMTU study not only demonstrated DMTU's ability to improve maize seed germination.." - are they not mutually exclusive?
Reviewer 3 Report
Thedraft “Effect of ROS...” by Li et al. (2023),has been assessed. Line 14, Reactive oxygen species (ROS) are an important part of component plant adaptation… Rewrite. Line 21, … which are staple cultivars grown in high-latitude cold regions of northeastern China. Line 23, … growth of the test maize seeds was dose dose-dependently. Line 25, .. water medium culturefor incubation. Line 29, .. normal culture...(??) Lines 29-30, … DMTU retarded the elongation of embryonic roots of testaffects maize seeds by regulating the production of endogenous H2O2. This sentence has not physiological sense. Neither did the last sentence. Together, the abstract needs to be rewritten. Introduction, although somewhat longer, is acceptable. Line 153, …. water (0 mmol·dm−3 DMTU)….(control)… or four dosases…. Lines 154-156, rewrite. Line 161, … O,…. Figs. 4E, F, it's not clear what the difference is between 200 mmol·dm-3 DMTU and DMTU/water. Explain. Lines 280-82, rewrite this sentence. Interesting results. Line 391, ….Spm spermine-induced H2O2 signaling. I'm missing a final paragraph that contains a general conclusion of this entire draft. Line 409, Seed material and germination. Regarding discussion, authors should keep in mind that there are a good number of references related to ROS and seed germination. Some of these references are very valid to put together a more scientifically sound discussion than the one presented. That is, authors need to work more on the scientific content of the discussion. Keep in mind that this work contains a good deal of experimentation which deserves to be properly exploited.Line 559, Plant Cell. Line 573, Plant J. Line 597, Crop J. Line 602, Plant Signal. Behav. Line 661, Phytochem.
Major:
- the abstract must be rewriten.
- the discussion should be more elaborated.
Minor:
- references should be reviewed.
- all errors, doubts and questions of the reviewer must be "point by point" answered.
Minor editing
Author Response
Responses to the Comments of the Reviewer
Thank you for reviewing our manuscript and for providing constructive comments and helpful suggestions, which helped us to improves the quality of the manuscript. We have revised the manuscript per the comments. Our responses to the comments are provided below( This revision is highlighted in green).
1) Line 14, Reactive oxygen species (ROS) are an important part of component plant adaptation… Rewrite
Response 1: Thank your for your suggestion. We have deleted the term "component" in line 15 in the revised manuscript.
2) Line 21, … which are staple cultivars grown in high-latitude cold regions of northeastern China.
Response 2: We agree with your suggestion. The phrase "which are... China" has been deleted in lines 21 in the revised manuscript.
3) Line 23, … growth of the test maize seeds was dose dose-dependently.
Response 3: We agree with your suggestion. We have deleted the phrase " of the test maize seeds" in line 23 in the revised manuscript.
4) Line 25, .. water medium culturefor incubation.
Response 4: Based on the suggestion, we have revised the expression in line 25 in revised manuscript.
5) Line 29, .. normal culture...(??)
Response 5: Thank you for your valuable comment. We have added the specific condition for normal culture, that is, 0 mmol·dm−3 DMTU (please see line 29 in revised manuscript).
6)Lines 29-30, … DMTU retarded the elongation of embryonic roots of testaffects maize seeds by regulating the production of endogenous H2O2. This sentence has not physiological sense. Neither did the last sentence. Together, the abstract needs to be rewritten. Introduction, although somewhat longer, is acceptable.
Response 6: We agree with the suggestion. In accordance with the suggestion, we have revised some statements in the Abstract and deleted the sentence in lines 29-31, that is, "…DMTU retarded the elongation of embryonic roots of testaffects maize seeds by regulating the production of endogenous H2O2".
7)Line 153, …. water (0 mmol·dm−3 DMTU)….(control)… or four dosases….
Response 7: We agree with the suggestion. We have changed “0 mmol·dm−3 DMTU” to “control”, and “numerous” to “four” (please see line 154 in the revised manuscript).
8) Lines 154-156, rewrite.
Response 8: Thank you for the valuable suggestion. We have revised the indicated description. Please see lines 155-159 in the revised manuscript.
9) Line 161, … O,…. Figs. 4E, F, it's not clear what the difference is between 200 mmol·dm-3 DMTU and DMTU/water. Explain.
Response 9: We agree with the suggestion. We have added an explanation of "DMTU/Water" in the legend of Figure 4 (lines 259-260 in the revised manuscript).
10) Lines 280-82, rewrite this sentence. Interesting results.
Response 10: Thank you for the valuable suggestion. We have revised the indicated description. Please see lines 284-286 in the revised manuscript.
11) Line 391, ….Spm spermine-induced H2O2 signaling. I'm missing a final paragraph that contains a general conclusion of this entire draft.
Response 11: We agree with the suggestion. We believe that the previously cited article is inappropriate for discussing the downregulation of H2O2 signaling by DMTU, which affects the associated antioxidant enzyme activities. Therefore, we have cited a more appropriate article and re-elaborated the discussion in lines 394–397; furthermore, we have revised the sentence in lines 556–559 in the Conclusion. The in-text citations have been renumbered accordingly. Please see lines 708–709 in the revised manuscript for more details.
12) Line 409, Seed material and germination.
Response 12: We agree with the suggestion. We have deleted "material and" in line 437 in the revised manuscript.
13) Regarding discussion, authors should keep in mind that there are a good number of references related to ROS and seed germination. Some of these references are very valid to put together a more scientifically sound discussion than the one presented. That is, authors need to work more on the scientific content of the discussion. Keep in mind that this work contains a good deal of experimentation which deserves to be properly exploited.
Response 13: We agree with the suggestion. In the Discussion, we have highlighted that DMTU reduces the viability of maize seed embryos and radicles, by comparing the results of this study with those of previous studies. Furthermore, we have improved the discussion on the involvement of DMTU in regulating H2O2 signaling and downstream related antioxidant enzyme activities; the in-text citations have been renumbered accordingly. Please see lines 364–377 and 417–421 in the revised manuscript for details.
14) Line 559, Plant Cell
Response 14: Thank you for pointing this out. We have changed "The Plant Cell" to "Plant Cell"; please see line 588 in the revised manuscript.
15) Line 573, Plant J.
Response 15: We agree with the suggestion. We have changed "The Plant Journal" to "Plant J."; please see line 602 in the revised manuscript.
16) Line 597, Crop J.
Response 16: Thank you for pointing this out. We have changed "The Crop Journal" to " Crop J."; please see line 625 in the revised manuscript.
17) Line 602, Plant Signal. Behav.
Response 17: Thank you for pointing this out. We have changed "Plant Signaling Behav" to "Plant Signal. Behav."; please see line 630-631 of the revised manuscript.
18) Line 661, Phytochem.
Response 18: Thank you for pointing this out. We have changed "Plant Signaling Behav" to "Plant Signal. Behav."; please see line 690 in the revised manuscript.
Round 2
Reviewer 1 Report
The authors have addressed all the issues in the MS, I have no further queries.
Author Response
Many thanks to the reviewers for recognising this manuscript.
Reviewer 2 Report
The authors responded satisfactorily to all my remarks and comments.
Author Response

(The authors gave the same response as above.)
